# TRAILS: Tree reconstruction of ancestry using incomplete lineage sorting

Iker Rivas-González[1]*, Mikkel H. Schierup[1], John Wakeley[2], Asger Hobolth[3]

**1** Bioinformatics Research Center (BiRC), Aarhus University, Aarhus, Denmark, **2** Department of Organismic and Evolutionary Biology, Harvard University, Massachusetts, United States of America, **3** Department of Mathematics, Aarhus University, Aarhus, Denmark

* iker_rivas_gonzalez@eva.mpg.de

**Data Availability Statement:** The python package for TRAILS can be downloaded and installed from pip (https://pypi.org/project/trails-rivasiker/), and the source code can be browsed at https://github.com/rivasiker/trails. The code for reproducing the

## Abstract

Genome-wide genealogies of multiple species carry detailed information about demographic and selection processes on individual branches of the phylogeny. Here, we introduce TRAILS, a hidden Markov model that accurately infers time-resolved population genetics parameters, such as ancestral effective population sizes and speciation times, for ancestral branches using a multi-species alignment of three species and an outgroup. TRAILS leverages the information contained in incomplete lineage sorting fragments by modelling genealogies along the genome as rooted three-leaved trees, each with a topology and two coalescent events happening in discretized time intervals within the phylogeny. Posterior decoding of the hidden Markov model can be used to infer the ancestral recombination graph for the alignment and details on demographic changes within a branch. Since TRAILS performs posterior decoding at the base-pair level, genome-wide scans based on the posterior probabilities can be devised to detect deviations from neutrality. Using TRAILS on a human-chimp-gorilla-orangutan alignment, we recover speciation parameters and extract information about the topology and coalescent times at high resolution.

## Author summary

DNA sequences can be compared to reconstruct the evolutionary history of different species. While the ancestral history is usually represented by a single phylogenetic tree, speciation is a more complex process, and, due to the effect of recombination, different parts of the genome might follow different genealogies. For example, even though humans are more closely related to chimps than to gorillas, around 15% of our genome is more similar to the gorilla genome than to the chimp one. Even for those parts of the genome that do follow the same human-chimp topology, we might encounter a last common ancestor at different time points in the past for different genomic fragments. Here, we present TRAILS, a new framework that utilizes the information contained in all these genealogies to reconstruct the speciation process. TRAILS infers unbiased estimates of the speciation times and the ancestral effective population sizes, improving the accuracy when compared to previous methods. TRAILS also reconstructs the genealogy at the highest resolution, inferring, for example, when common ancestry was found for different parts of the

figures in the manuscript can be found at https://github.com/rivasiker/trails_paper.

**Funding:** This work was supported by the Novo Nordisk Foundation (NNF18OC0031004 to MHS) and the Independent Research Fund Denmark, Natural Sciences (6108-00385 to MHS). The funders had no role in study design, data collection and analysis, decision to publish, or preparation of the manuscript.

**Competing interests:** The authors have declared that no competing interests exist.

genome. This information can also be used to detect deviations from neutrality, effectively inferring natural selection that happened millions of years ago. We validate the method using extensive simulations, and we apply TRAILS to a human-chimp-gorilla multiple genome alignment, from where we recover speciation parameters that are in good agreement with previous estimates.

## Introduction

Orthologous sites in two or more sequences share a unique genealogical history, with coalescent events happening at certain time points in the past. In the absence of recombination, all sites along the sequences follow the same genealogy. In reality, however, ancestral recombination events might have decoupled consecutive sites, generating an array of segments with different yet correlated genealogies, collectively known as the ancestral recombination graph (ARG) [1, 2]. In principle, if inferred accurately, the ARG contains all available information about the demography of the samples, and it can be used to estimate population parameters (such as the recombination rate and the ancestral effective population sizes), historical events (such as introgression and hybridization), and selective processes [3]. The ARG, however, is challenging to infer because the underlying genealogies along the genome alignment cannot be directly observed. Instead, inference of the genealogy along the genome relies on the site patterns of the accumulated mutations.

The ARG can also be formulated as a spatial process along the genomic alignment [4]. This process, however, contains a long-range correlation structure because if two recombination events happen flanking a genomic fragment, the fragment might be surrounded by the exact same genealogy. However, disregarding the fact that the process is non-Markovian in nature, the ARG can be approximated by a hidden Markov model (HMM), where the genealogy of a certain genomic position only depends on the genealogy of the previous position [5, 6]. It has been shown that this approach, commonly referred to as sequentially Markovian coalescent or SMC, is a good approximation of the true coalescent-with-recombination process [7]. Perhaps the simplest of such models is the pairwise sequentially Markovian coalescent (PSMC) [8], in which the ARG between two sequences (typically, the two copies of a diploid individual) is modelled. Here, the hidden states are coalescent events that happen in discretized time intervals, which correspond to two-leafed gene trees (Fig 1A). The transition probabilities between pairs of hidden states can be calculated using standard coalescent theory, parameterized by the recombination rate and the ancestral effective population sizes ($N_e$) in each time interval [8]. PSMC, and other SMCs, such as MSMC [9], MSMC2 [10], ASMC [11], and SMC++ [12], allow the use of standard HMM machinery to infer population parameters, and are thus also useful for inferring the most plausible coalescent times from the posterior decoding. However, SMC models are generally restricted to a single coalescent event between a pair of samples, which limits their usefulness. More recently, there have been new developments to model multiple samples explicitly. For example, ARGweaver [13], Relate [14], tsinfer+tsdate [15, 16] or ARG-Needle [17] use techniques such as resampling, threading and mathematical approximations to sequentially build the ARG [18].

These models are typically used to analyze samples from the same species to get within-species information about the ancestral process. Analyzing inter-species coalescent events adds another layer of complexity, since the coalescent events need to be contained within the underlying phylogeny or speciation tree [22–24]. Moreover, the models described above typically use the presence or absence of a certain mutation to construct haplotypes, but ignore or filter out instances where more than two alleles are observed. This infinite sites model poses a

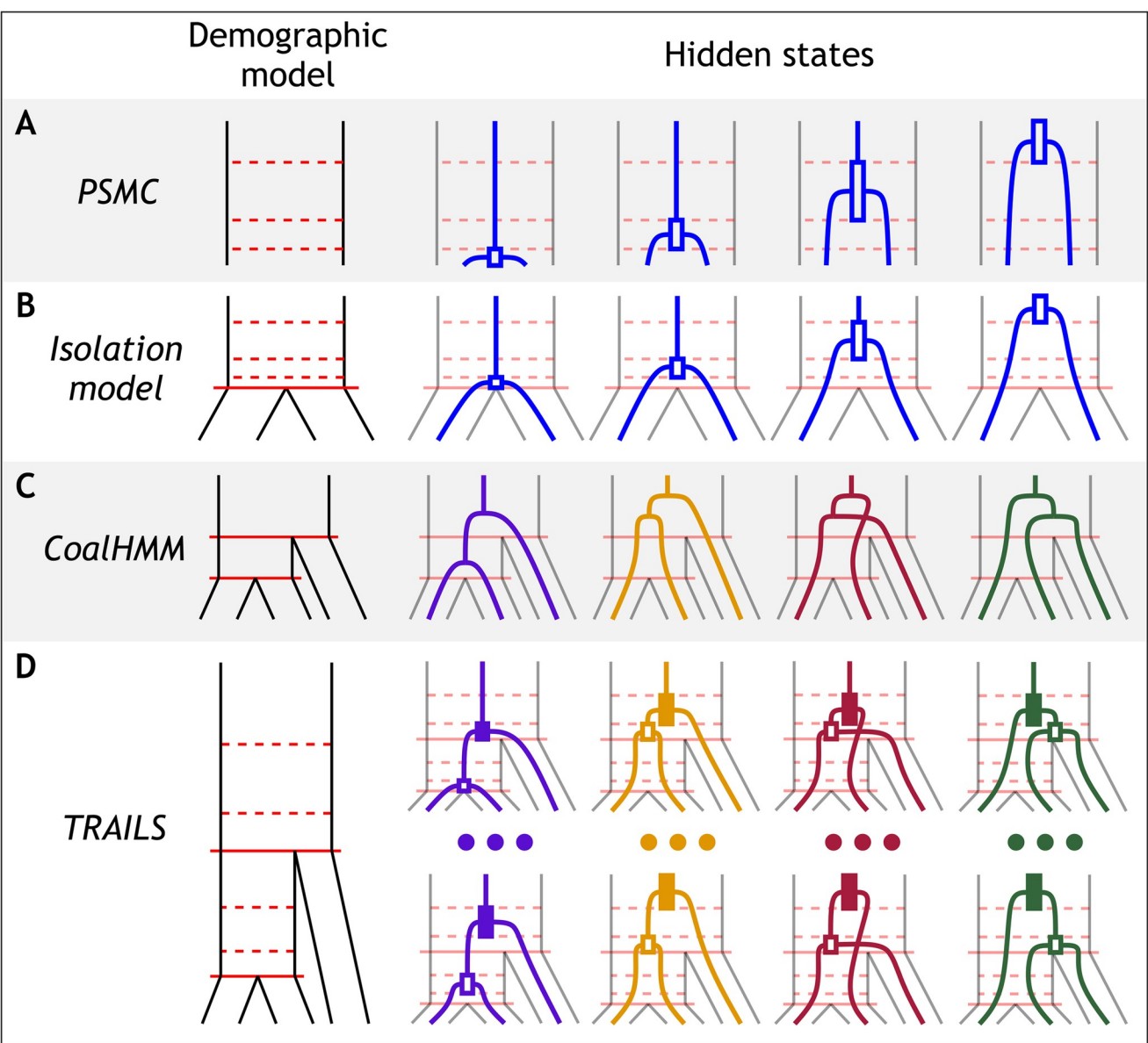

**Fig 1. TRAILS is a HMM that reconstructs the time-resolved multi-species ARG for three genomes.** TRAILS extends the isolation model (B) [19] to three species, by combining the time discretization of PSMC-like models (A) [8] with the topologically aware hidden states of CoalHMM (C) [20, 21]. The resulting hidden states of TRAILS (D) are three-leaved genealogies with two discretized coalescent events and one of four possible topologies. A full list of the 27 possible hidden states when $n_{AB} = n_{ABC} = 3$ can be consulted in Fig K in S1 Text.

problem for inter-species analysis, because recurrent mutation is more likely to happen, generating instances of sites that have experienced more than a single mutation [25, 26].

Some other models have tried to extend these concepts for the analyses of multiple species. For example, the coalescent-with-isolation model [19] is conceptually similar to PSMC, but, backwards in time, the two analyzed samples are kept isolated until the speciation event, after which they can coalesce (Fig 1B). This model can be used to estimate the speciation time between the two samples and the $N_e$ of the ancestral species, and an extension of it can be used to model isolation-with-migration [27]. These models, similar to SMCs, can output a posterior decoding of the coalescent times.

Beyond two samples, CoalHMM models the coalescent with recombination of three species [20, 21], where the hidden states are the four possible genealogies that might arise within the underlying species tree (Fig 1C). Two of the four genealogies differ from the species tree, which generate incongruencies that might pose a problem for standard phylogenetic reconstruction. Nevertheless, this phenomenon, commonly referred to as incomplete lineage sorting or ILS, is very informative about the demographic parameters of the underlying species tree, and CoalHMM can thus be used to estimate ancestral $N_e$ and two speciation times. Moreover, CoalHMM uses a substitution model for mutations, so recurrent mutations are allowed. However, unlike SMCs, CoalHMM does not model coalescent events at discretized time intervals and, instead, coalescent times are modelled as single time points within an individual branch. Because of this, some of the parameter estimates of CoalHMM are biased [21], and, although obtaining accurate estimates is still possible [28], the debiasing procedure involves costly coalescent simulations. Moreover, posterior decoding can only be performed on the topology of the gene trees, and not on the coalescent times.

Here we present TRAILS, an HMM that combines modelling the information-rich ILS signal in the style of CoalHMM and the time discretization of SMC-like models to infer unbiased estimates of the demographic parameters (ancestral $N_e$ and speciation times), and to enable the posterior decoding of both topology and coalescent times. In TRAILS, the hidden states are three-leaved gene trees, each with a specified topology and two coalescent events that happen at discretized time intervals on an underlying speciation tree (Fig 1D and Fig K in S1 Text). The genealogies are rooted by a fourth sample from an outgroup species. The transition probabilities between the hidden states of TRAILS are calculated using coalescent-with-recombination theory for one, two and three lineages that segregate within the branches of the phylogeny. We provide formulas in matrix notation to calculate these transition probabilities for a varying number of discretized time intervals (see Methods for a short explanation, and S1 Text for an in-depth description of the theory). The emitted states are sites in a four-way multiple genome alignment, containing the sequences of the three species and the outgroup. The transition and emission probabilities are parameterized by two ancestral $N_e$, speciation times, and the recombination rate. Keeping the mutation rate at a fixed value, TRAILS allows for the estimation of the other parameters by optimizing the HMM likelihood given the alignment. After fitting the HMM, TRAILS can perform posterior decoding of the hidden states, inferring a posterior probability of coalescent events through time within the speciation tree.

Here we derive the transition and emission probabilities, implement the model and demonstrate its use on simulated and real data. After optimizing the population parameters using TRAILS on a simulated dataset, we show that increasing the number of discrete coalescence intervals reduces the bias in the parameter estimation. We also show how the posterior decoding can accurately reconstruct the true ARG, by inferring the topology of gene trees and the time in which coalescent events occurred. We perform additional simulations to show that the posterior decoding of TRAILS can be used to detect selective sweeps that happened on ancestral branches of the phylogeny. Finally, we analyze a human-chimp-gorilla-orangutan alignment, inferring the demographic parameters of the underlying species tree and performing genome-wide posterior decoding at the base-pair level.

## Results

### Parameter estimation

The transition and emission probabilities of coalescent hidden Markov models (HMMs) are parameterized by the demographic model, i.e., by the speciation times, ancestral effective population sizes ($N_e$) and recombination rate ($\rho$). This means that using standard HMM

algorithms, these parameters can be optimized to obtain the model that best explains the observed data. Both in CoalHMM and in TRAILS, numerical optimization is performed on the log-likelihood calculated using the forward algorithm, given a four-way genome alignment of three focal species and an outgroup. The maximum likelihood estimates of the demographic model are then found using a bound-constrained search algorithm that optimizes the likelihood function by evaluating it directly.

Previous work using coalescent HMMs has shown that the estimation of the demographic parameters is challenging. In CoalHMM, for example, the parameter estimates are highly biased [21], especially for the ancestral $N_e$ and the recombination rate. It is possible to obtain close to unbiased estimates, but this requires a costly simulation procedure [28]. The source of the bias seems to be the restrictive state space of CoalHMM [21], which includes the topology of the genealogy but no information on when the coalescents happened within each branch of the tree (Fig 1C).

TRAILS overcomes this issue by extending the state space to include coalescent events that can happen in discretized time intervals. To demonstrate that TRAILS can perform unbiased parameter estimation, we generated twenty 10-Mb four-way alignments using msprime [29] by choosing a demographic model similar to the human-chimp-gorilla-orangutan speciation tree (see Methods, Simulations for details). The simulated sequence alignments were analyzed using TRAILS, estimating the times, ancestral $N_e$ values and recombination rate depicted in Fig 2A. Parameters were estimated for $n_{AB} = n_{ABC} = 1$ and $n_{AB} = n_{ABC} = 5$, where $n_{AB}$ is the number of intervals between speciation events and $n_{ABC}$ is the number of intervals deep in time, in the common ancestor of all three species.

In the model where $n_{AB} = n_{ABC} = 1$, which is equivalent to the original CoalHMM model, parameter estimates deviate from their true values in the simulations, especially for $t_2$, $N_{AB}$ and

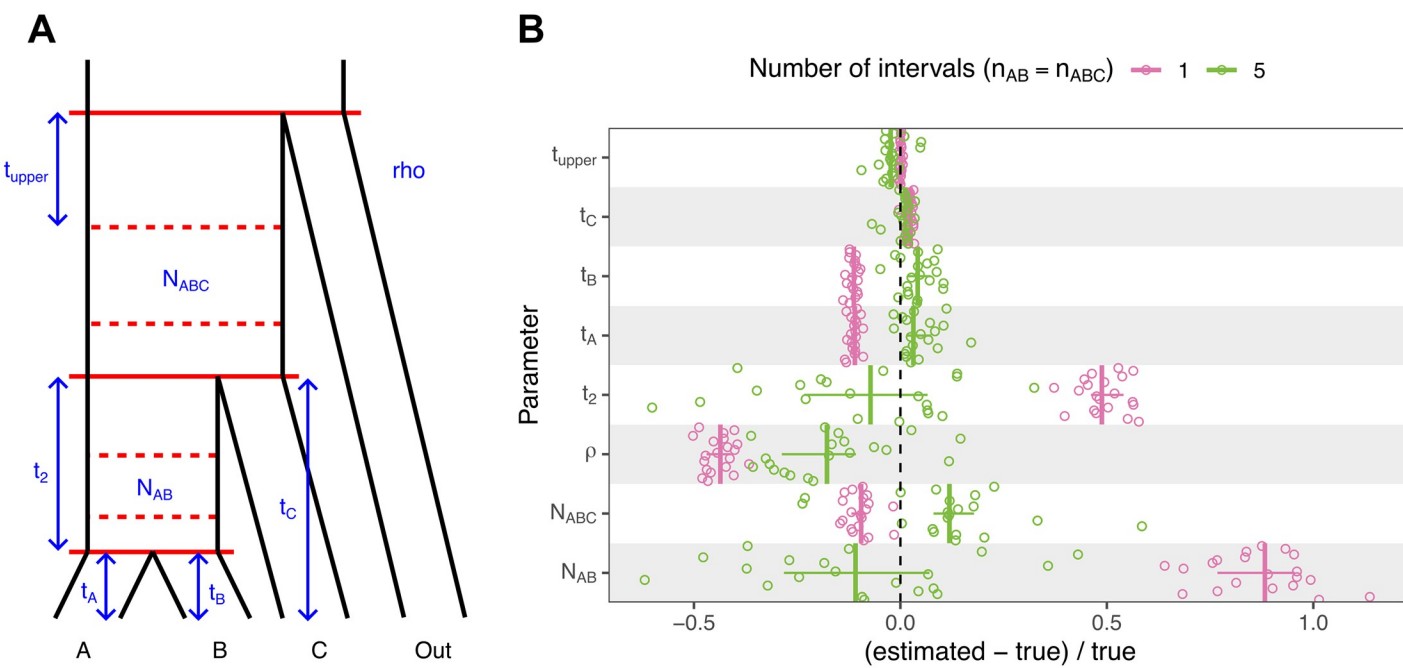

**Fig 2. Increasing the complexity of TRAILS reduces the bias of the estimated parameters.** (A) Diagram (not to scale) of the demographic model with all the optimized parameters in blue for the non-ultrametric case. In an ultrametric model, $t_1$ would correspond to the time from present to the shallowest speciation event, where $t_1 = t_A = t_B = t_C - t_2$. (B) Relative error of parameter values estimated from 20 simulated msprime genomes for $n_{AB} = n_{ABC} = 1$ (in pink) and $n_{AB} = n_{ABC} = 5$ (in green). Each independent run corresponds to a dot, vertical lines are median values, and vertical lines correspond to interquartile ranges. Units are normalized as (estimated-true)/true to ease comparison across parameters.

$\rho$ ([Fig 2B], in pink). Using a larger number of intervals ($n_{AB} = n_{ABC} = 5$) improves the accuracy of the parameter estimation ([Fig 2B], in green). An exception to this is the recombination rate, which is still underestimated (albeit less so), possibly due to recombination events that produce small changes in the coalescent tree (e.g., not changing in the topology and only moving the coalescent times by a few generations) and are thus missed from the sequence data. Generally, however, the simulation results demonstrate that the source of the bias in the parameter estimation in CoalHMM can be alleviated with a more flexible model that includes coalescent times at discretized time intervals.

## Posterior decoding of simulated data

Posterior decoding using the parameters estimated from the alignment can be performed using the transition and emission probabilities computed by TRAILS for a specific demographic model. In contrast to other coalescent-based HMMs, the resulting posterior probabilities are, however, hard to visualize, since each hidden state will have its own topology, and first (or more recent) and second (or more ancient) coalescent time intervals ([Fig 1D] and Fig K in [S1 Text]). To overcome this, we summarize the posterior probabilities by grouping states that share certain features. For example, the posterior probabilities of all states that share the same topology can be summed. Similarly, the posteriors of all states with the same first or second coalescent times can also be summed.

In order to have a ground truth for comparison, the posterior decoding was performed on 100 kb of an alignment simulated using msprime, with a demographic model identical to that used in [Fig 2B] (see [Methods] for a full description of the model). The resulting posteriors can capture the true topology and the second coalescent time quite accurately ([Fig 3A and 3B],

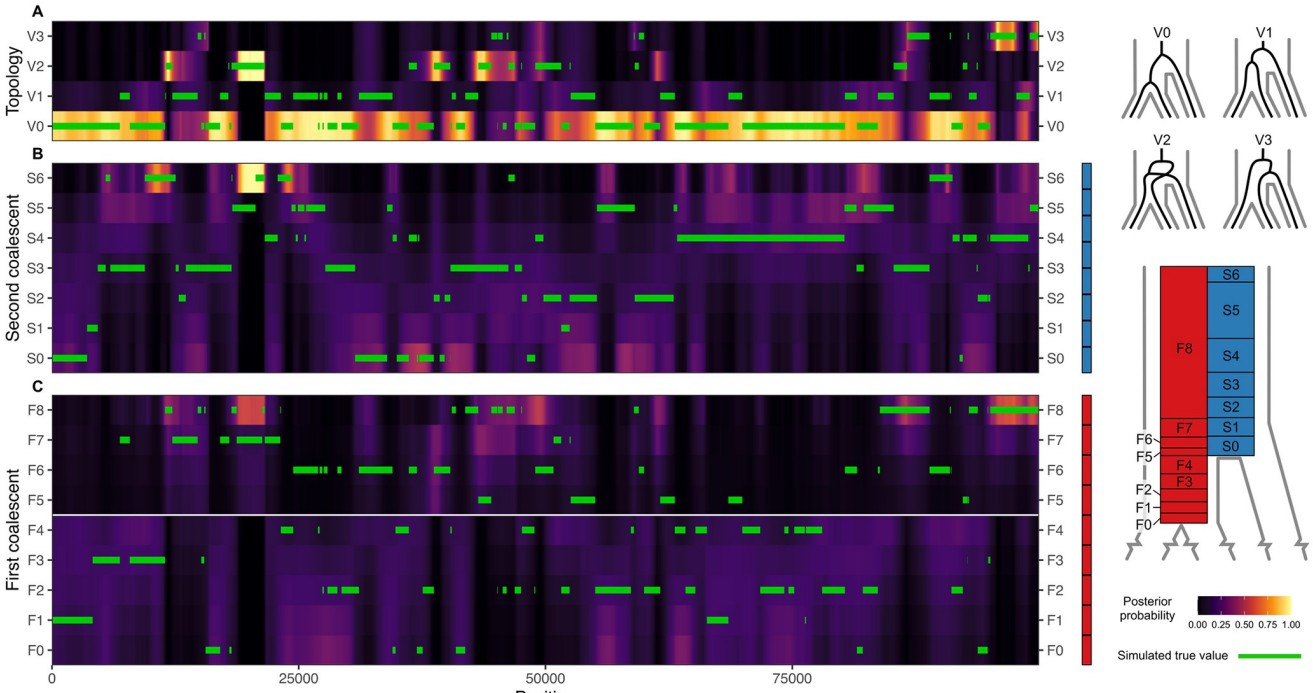

**Fig 3. Posterior probability for the topology (A), second coalescent event (B) and first coalescent event (C) of a 100 kb msprime simulation.** "First" and "second" refer to the order in which coalescent events happen, backwards in time. The true empirical topology and coalescent times are plotted as green lines.

respectively), while the first coalescent time (Fig 3C) is harder to estimate. Additionally, V1 segments (Fig 3A) are easily misclassified as V0 segments, since V0 and V1 only differ in branch lengths but not in topology (see Fig 3 and Fig I in S1 Text), and, thus, the emitted site patterns for V0 and V1 are similar.

## Posterior decoding from simulated data with selection

To showcase how the posterior decoding could be useful to infer deviations from the standard coalescent, we simulated a 200 kb alignment using SLiM [30] containing a single positively selected variant ($2N_e s = 175$) in position 100 kb of the simulated alignment that arises at the first interval backwards in time, where the first two species merge in the speciation tree (interval S0 in Fig 4A). The site is strongly positively selected, but it lies well within the possible range of values for selection coefficients recorded in humans, with the lactase gene having up to $2N_e s = 1000$ in some human populations [31]. The demographic parameters were the same as those used for Figs 2 and 3.

Both the true empirical values and the posterior decoding show that there is an overrepresentation of second coalescent events happening in interval S0 (Fig 4A), which is qualitatively different from the neutral case (Fig 3B). The positively selected variant confers a big advantage and is fixed rapidly in the population, with an expected fixation time of

$$\frac{4}{s}\left(\ln(2N_e s) + \gamma - \frac{1}{2N_e s}\right) = 2622 \text{ generations or 65,550 years for an } N_e = 10,000 \text{ and a genera-}$$

tion time of $g = 25$ years, where $\gamma \approx 0.577$ is Euler's constant [32]. In contrast, for a neutrally

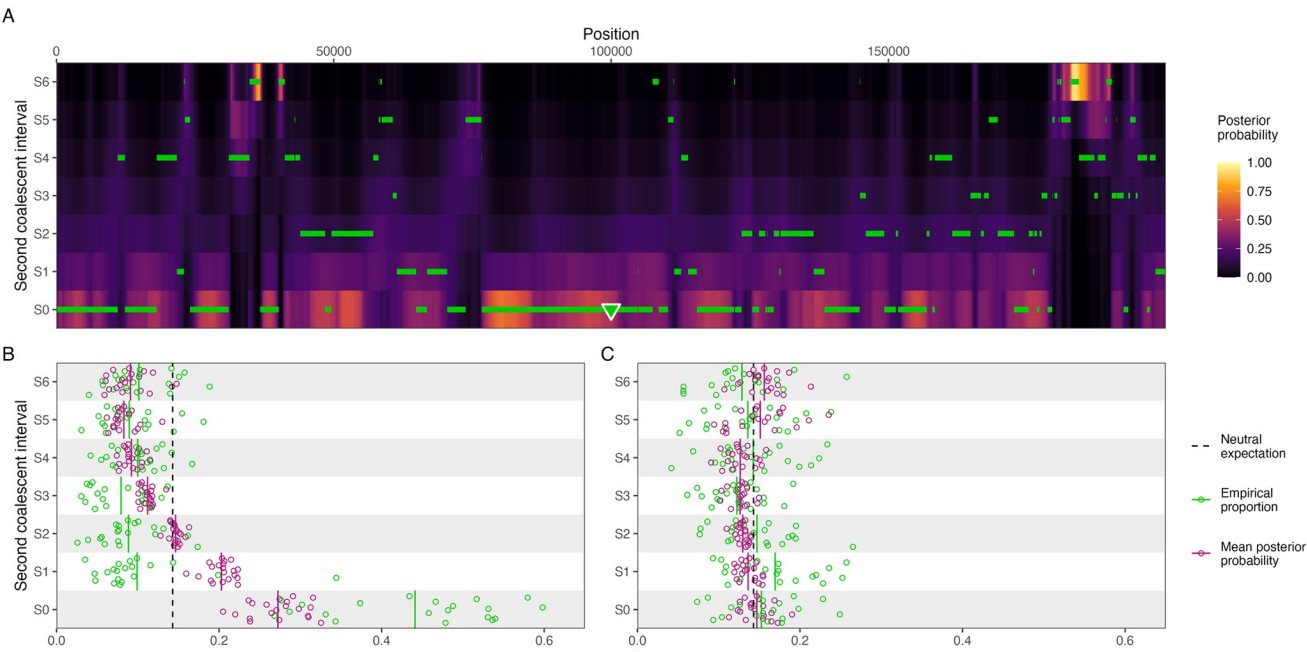

**Fig 4. The posterior probability can be used to detect deviations from the neutral expectation.** (A) Posterior probability of the second coalescent event for a simulated 200 kb region containing a positively selected variant at position 100 kb that arises in interval S0, represented by a triangle. The true simulated coalescent times are plotted as green horizontal lines. (B) Mean posterior probability for each second coalescent interval (purple), and the empirical true proportion of sites for each interval (green) for 20 simulated replicates with a selective sweep, using the same model as in (A). The theoretical neutral expectation is plotted as a black dashed line, and time intervals are adjusted so that all intervals have equal probability of observing a coalescent event. Continuous vertical lines represent mean values of the simulations. (C) Same as in (B), but for a neutrally evolving region, using the same model as in Fig 3.

evolving site, the expected fixation time is $4N_e = 40,000$ generations or 1 million years [33]. The effect of such strong selective force is that whatever polymorphism existed at the selected locus is quickly purged from the population, and, with it, linked neutral variants are hitchhiking along, causing a selective sweep. As a result, there is an excess of coalescent events happening in interval S0, which can be discerned from the posterior probabilities.

The signal observed in the posterior decoding can be summarized by computing the mean posterior probability per time interval and comparing it to the theoretical neutral expectation. There is a clear excess of coalescent events estimated to happen at the interval where the beneficial mutation arises (S0 in Fig 4B), although there is also an excess of coalescent events inferred by the posterior in nearby intervals (S1 and S2), where coalescents are misclassified due to close proximity in time. In any case, the pattern observed for the selective sweep in Fig 4B is in stark contrast with the neutral case shown in Fig 4C, where the posterior falls within the expected values. This demonstrates that deviations from neutrality can be inferred using the posterior decoding of TRAILS, and one could devise a windowed genome-wide scan for selection by summarizing the posterior as proposed in Fig 4B and 4C.

## Parameter estimation from a HCGO alignment

ILS happens pervasively on the branches of the tree of life, spanning taxonomically diverse groups such as marsupials [34], birds [35, 36], fishes [37], plants [38], and mammals [39, 40], including primates [28, 41, 42]. For example, there is around 32% of ILS in the human-chimp ancestor, with 16% of the genome grouping human and gorilla, and another 16% grouping chimp and gorilla [28]. These estimates were obtained using CoalHMM [21], together with estimates for ancestral $N_e$ and split times, which were debiased using simulations. Here, we apply TRAILS to a 50 Mb human-chimp-gorilla alignment from chromosome 1 with orangutan as an outgroup to infer population genetics parameters and to gain information about the coalescent times and the topology through posterior decoding.

Using MafFilter [43], the alignment was first preprocessed to extract the species of interest (human, chimp, gorilla and orangutan), to merge consecutive alignment blocks, and to filter out small blocks (see Methods for further details). Using `biopython` [44], 50 Mb were extracted from chr1, namely the region from 25 Mb to 75 Mb. This region was used as the input for TRAILS, choosing the parameter values estimated in Rivas-González et al. [28] as starting values for the optimized parameters, setting $n_{AB} = n_{ABC} = 3$, and using the L-BFGS-B algorithm for model fitting [45, 46], although other bound-constrained method can also be used. To get more accurate parameter estimates, the optimized parameters were used as starting parameters for a second TRAILS run, where $n_{AB} = 3$ and $n_{ABC} = 5$.

The resulting estimates are displayed in Fig 5A. Assuming a mutation rate of $\mu = 1.25 \times 10^{-8}$ per site per generation and a generation time of $g = 25$ years [47, 48], the speciation time estimates are in good agreement with previously inferred values. Using the human tip branch length, we estimate the time until the HC split at 5.51 million years ago (95% CI: [5.43, 5.54], $\sim 4$–7 MYA from literature [21, 28, 49–51]), the HCG split at 10.40 MYA (95% CI: [10.27, 10.40], $\sim 8$–12 from literature [21, 28, 50, 51]), and the HCGO split at 18.55 MYA (95% CI: [18.37, 18.73], $\sim 10$–20 from literature [28, 41, 50]). Moreover, ancestral $N_e$ inferred for the HC ancestor (167,400, 95% CI: [165, 548, 170, 361]) and for the HCG ancestor (101,290, 95% CI: [100, 467, 101, 492]) are consistent with previous estimates using CoalHMM (177,368 and 106,702, respectively [28]). Using the estimates for $t_2$ (in generations) and $N_{AB}$, we get a probability of ILS equal to

$$\text{ILS} = \frac{2}{3}\exp(-t_2/(2N_{AB})) = \frac{2}{3}\exp(-195,000/(2 \times 167,400)) = 37\%,$$

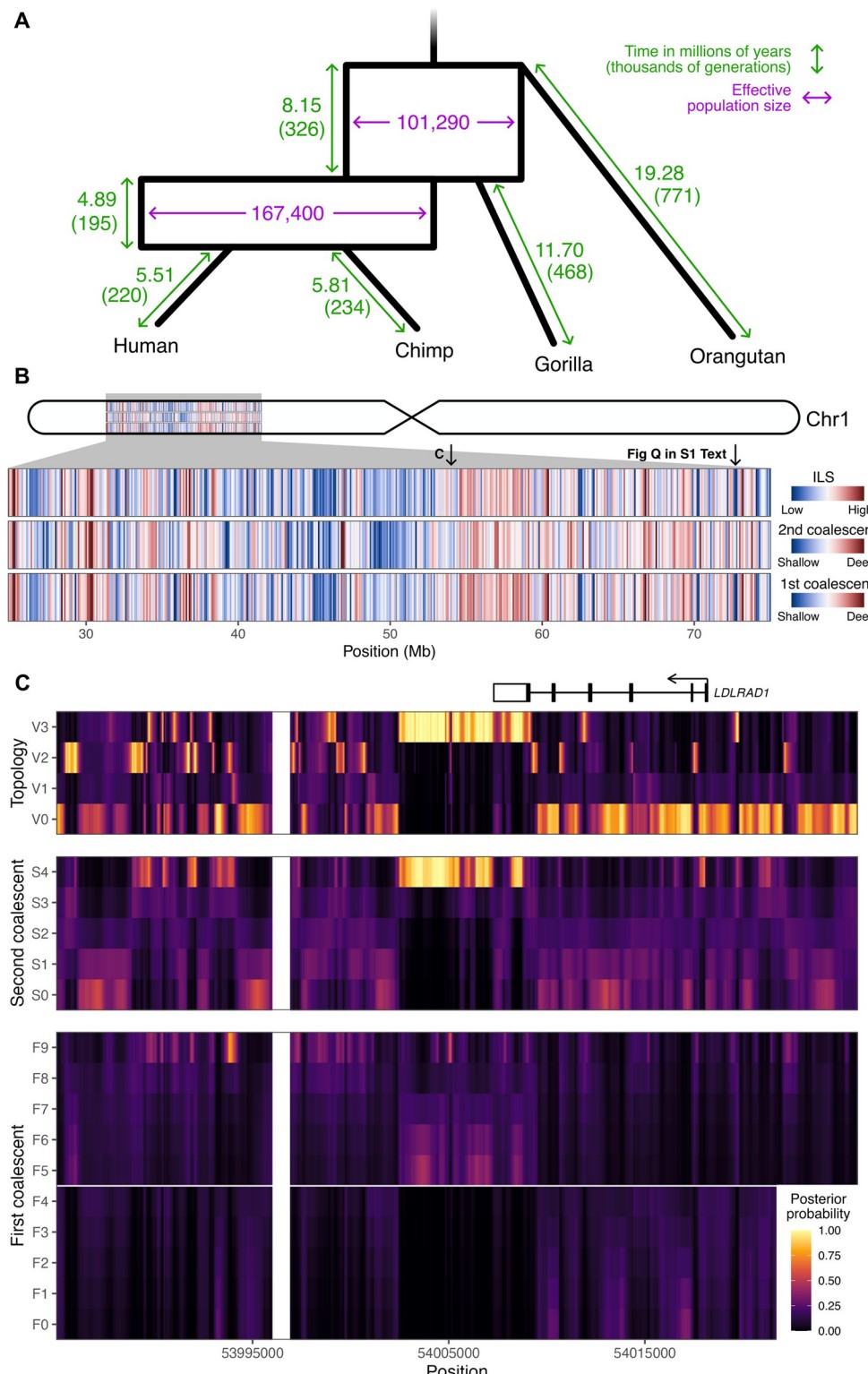

**Fig 5. TRAILS output for 50 Mb of chromosome 1 of the HCGO alignment.** (A) Estimates for the speciation times (green) and ancestral $N_e$ (purple) of the speciation process, optimized using TRAILS and assuming a mutation rate of $\mu$ = $1.25 \times 10^{-8}$ per site per generation. To convert time from generations to millions of years, a generation time of $g$ = 25 years per generation was used. (B) Genome-wide variation of ILS, and first and second coalescent times. (C) Posterior decoding of the topology, and first and second coalescent events for a zoomed-in region in chromosome 1. As in Fig 3,

both V0 and V1 correspond to the species topology (((H,C),G),O);, V2 corresponds to (((H,G),C),O);, and V3 to (((C, G),H),O);. The *LDLRAD1* gene is plotted on top, where exons are represented as boxes, coding regions as filled boxes, and introns as horizontal lines.

so our parameter estimates suggest the ((human, gorilla), chimp) topology in 18.5% of the genome and the ((chimp, gorilla), human) topology in 18.5% of the genome. Finally, the recombination rate was estimated to be $\rho = 1.19 \times 10^{-8}$ per site per generation, which matches the rate estimated for present-day humans [52].

TRAILS allows for the independent estimation of each individual branch length, which is useful for non-ultrametric trees. Fig 5A shows that the branch leading to chimps is longer than that leading to humans by around 5.9%, and the gorilla branch is longer than both the human (12.6%) and the chimp (9.1%) branches (calculated from the second speciation event to present). This deviation from the molecular clock is well supported by previous studies [53], and is likely because of different branches accumulating a different number of mutations per year, either due to an acceleration or deceleration of the mutational process, changes in the average time of reproduction, or a combination of these [54–56].

In summary, we have demonstrated that TRAILS is able to infer demographic parameters that are in agreement with estimates from the literature. More importantly, it does so without the need for any post-processing or corrections, avoiding the use of fossil calibrations [50] or debiasing procedures [21, 28].

## Posterior decoding of the HCGO alignment

Posterior decoding was then performed using the optimized parameters and setting $n_{AB} = n_{ABC} = 5$. In order to get an understanding of the genome-wide variation of ILS and coalescent times, the resulting posterior probabilities were summarized in 100 kb windows along the chr1 region in three different ways. First, the mean posterior probability was calculated for each of the four possible topologies, by first summing the posteriors of all hidden states sharing the same topology for each site, and then averaging over all sites on the 100 kb window. The resulting probabilities were then used to calculate a proxy for ILS, by summing the probabilities of observing the ILS topologies (V2 and V3 in Fig 3). Second, using a similar procedure, the mean posterior probability was calculated for each of the six possible intervals for the first coalescent event, by first summing the posteriors of all hidden states sharing the same first coalescent interval for each site, and then averaging over all sites in the 100 kb window. As a proxy for the first coalescent time, integers from 1 to 6 were assigned to each interval in chronological order backwards in time, and a weighted mean of those integers was computed, where the weights were the mean posterior probabilities per window. Third, the same quantity as for the first coalescent was computed for the second coalescent.

After filtering outliers smaller than the 1st percentile and larger than the 99th percentile, these proxies were plotted as heatmaps (Fig 5B). The first coalescent and the ILS proportion show a very strong correlation ($\rho = 0.979$), likely reflecting that V2 and V3 can only happen in the common ancestor of all three species, so when ILS is present, coalescent times are generally deeper (and vice versa). This signal is also captured, although more weakly, by the correlation between the second coalescent and the ILS proportion ($\rho = 0.483$). This can be explained by knowing that, conditional on ILS, the second coalescent follows a convolution of exponentials of rates 3 and 1 [57], while, conditional on V0 (i.e., conditional on the first coalescent happening between speciation events), the second coalescent simply follows an exponential or rate 1.

Thus, if more ILS is present in a certain window, then, on average, the second coalescent will tend to happen deeper in time.

Fig 5B also shows how, at the 100 kb level, the genome displays spatial covariation in the amount of ILS and the time to coalescence that exceeds stochastic effects of a neutral coalescent process. This is in line with previous results [28], where ILS proportions are affected by genomic features such as gene density, recombination rate, and the effects of linked selection.

A zoomed-in region of around 41 kb is shown in Fig 5C, which shows a long fragment of the ((chimp, gorilla), human) topology (V3). This fragment is unusually long, spanning 6,800 bp, and it is highly implausible following the demographic model inferred by TRAILS. Thanks to the posterior decoding of the coalescent times performed by TRAILS, we can observe that, for this fragment, the first coalescent event backwards in time happens close to the second speciation time (in interval F5), while the second coalescent event happens in the deepest time interval (S4).

One explanation for such a long V3 fragment is that it might be influenced by selection, which would maintain the alternative topology uninterrupted for a long period of time. Another explanation could be that this fragment is introgressed, especially given that the first coalescent event is shallow and that the fragment is long. For comparison, another region in chromosome 1, which also shows an excess of V3 topology, has a much more variable distribution of coalescent times, and it is more fragmented (Fig Q in S1 Text). Such detailed information about the timing of coalescent events is only possible thanks to the time discretization of TRAILS, and these details would have been missed, for example, in the posterior decoding of CoalHMM (recall Fig 1C).

The V3 fragment in Fig 5C overlaps with the last exon of the *LDLRAD1* gene, which codes for a lipoprotein receptor (UniProt: Q5T700). LDLRAD1 does not show signals of positive selection in hominids (based on dN/dS values from Rivas-González et al. [28]). Additionally, this gene is not particularly constrained in primates, as measured by PhastCons [58] and PhyloP [59], and it is not enriched for repeat elements, as retrieved from the UCSC Genome Browser [60]. While we were unable to point out a specific cause for the pattern observed for LDLRAD1, Fig 5C showcases how TRAILS can be used to infer the topology and coalescent times of protein-coding genes at the base-pair level across millions of years of evolution. Comparing the posterior decoding with genomic covariates can reveal selective processes affecting the sorting of lineages [28] or solve cases of phenotypic hemiplasy [34].

## Discussion

Coalescent-based approaches for analyzing genomic data are essential tools for understanding the ancestral history of species. Here, we have introduced TRAILS, an HMM that models the topology and the two coalescent events for gene genealogies within a phylogeny of three species. TRAILS can accurately infer population genetic parameters (ancestral $N_e$, speciation times and recombination rate). From the posterior decoding, the three-species ARG can be inferred at the base-pair level, providing insight into the ancestral history of the species at high resolution. Deviations from neutrality can be detected by summarizing the posterior decoding in windows and running genomic scans to find excess of coalescents happening at certain time intervals, such as proposed in Fig 4. Moreover, more coarse-grained summaries of the posterior decoding spanning several kilobases could be used to infer the genome-wide variation of ILS or coalescent times, potentially revealing correlations with other genomic features such as variation in the recombination rate or selection [28, 61].

As demonstrated here, the posterior decoding from TRAILS is a powerful way to infer details of the ARG in the context of speciation, together with departures from the neutral

expectation. Recurrent selective sweeps that have happened during the speciation process are hypothesized to be drivers of speciation, and to greatly influence the genealogical landscape of present-day genomes. For example, the human X chromosome contains long haplotypes shared across all non-African populations [62], spanning large genomic regions that are both lacking Neanderthal introgression [63], and showing very low rates of ILS in the human-chimp ancestor [64]. This suggests that the X chromosome has a unique evolutionary history which is greatly affected by gene flow (or lack thereof), and that these low-diversity regions might be related to genetic incompatibilities that arose during the speciation of ancestral hominids. TRAILS can help locate these ancient sweeps and infer when they occurred, potentially illuminating when and how genomes were affected by selection during the speciation process.

Using posterior decoding, regions that show unusually high levels of an alternative topology with very shallow coalescents can also be detected, which could indicate ancient introgression or hybridization events happening between ancestral branches of the species tree. Such ancient introgression events have been reported to be pervasive among some branches in the primate species tree [50], although they can be difficult to distinguish from ILS [65] unless explicitly modelled. TRAILS could be extended to model introgression more directly by including additional hidden states representing introgressed genomic fragments. These would have exceedingly short coalescent times compared to the deep coalescent ILS states [28], and TRAILS provides the mathematical framework to distinguish between these two cases.

TRAILS could also be extended to accommodate variation in $N_e$ along individual ancestral branches in the species tree, conceptually very similar to what is done in PSMC analyses from a single extant genome [8]. Modelling variation in $N_e$ can elucidate how speciation events might have happened. For example, population sizes that are maintained more constant during the speciation event might indicate a cleaner split, while increased ancestral $N_e$ just prior to the estimated time of speciation (here equalled to the total cessation of gene flow) might point to a period of elevated population structure and a prolonged species separation with migration [27]. Modelling changes in the demography around speciation events might also help us detect and characterize instances of complex speciation, as proposed, for example, by Patterson et al. [49].

The current implementation of TRAILS for calculating the transition probability matrix of the HMM is restricted to three species and a relatively few number of hidden states (see Fig P and section 9 in S1 Text for a discussion on the running times). With more efficient algorithms, future extensions of TRAILS could be devised to analyze more than three species, thus allowing for the inference of the speciation tree and the multi-species ARG for more taxa. Based on the parameters estimated by TRAILS for the HCGO alignment (Fig 5A), the proportion of ILS between humans (or chimps), gorillas and orangutans would be around 13%. While this violates one of the assumptions of TRAILS, which is that there should be inappreciable ILS between the outgroup and the rest of the analyzed species, it also showcases the need for models that are able to accommodate more species (see subsection 2.4 in S1 Text). This could help us resolve more complex patterns of ILS, which include phenomena such as anomaly zones [36, 66, 67].

## Methods

The transition probabilities between the hidden states of TRAILS can be calculated from a series of interconnected continuous-time Markov chains (CTMCs) that model the coalescent with recombination of two contiguous nucleotides for one, two or three sequences. The CTMCs are parameterized by the ancestral $N_e$, speciation times and recombination rate. The transitions for TRAILS are subsequently calculated by conditioning the CTMCs on the

topology and coalescent times of the gene trees at those two sites, binning coalescent events into discretized time intervals along the speciation process. Additionally, the emission probabilities for each hidden state are calculated from a CTMC of the mutational process by choosing a certain substitution model. In this section, we provide a summary of the model, and the full explanation can be consulted in S1 Text.

### Continuous-time Markov chains for the ancestral process

The coalescent with recombination between two sites can be approximated as a continuous-time Markov chain (CTMC). For one sequence, the left and the right sites can be either linked or unlinked, so there are only two possible states for the CTMC. Two linked sites become unlinked when a recombination event happens between them, which happens with a rate of $\rho_1$. On the other hand, the unlinked left and the right sites become linked when a coalescent event happens between them, with a rate of $\gamma_1$. These two transitions can be gathered in a transition rate matrix

$$\boldsymbol{Q_1} = \begin{pmatrix} -\gamma_1 & \gamma_1 \\ \rho_1 & -\rho_1 \end{pmatrix}. \tag{1}$$

From this transition rate matrix, we can calculate the probability matrix $P_A$ as $\exp(t\boldsymbol{Q_1})$, which gives the probability of the sites being unlinked or linked at a certain time $t$ given that the chain starts in the unlinked state (first row) or unlinked state (second row).

When two sequences are involved, the state-space of the CTMC becomes more complex. Apart from the coalescent and the recombination events described above, sites can also coalesce irreversibly backwards in time with rate $\gamma_2$, which happens when two left (or two right) sites from two different sequences find common ancestry. The resulting rate matrix (Fig E in S1 Text) for the coalescent with recombination with two sequences then corresponds to a CTMC with 15 states (Fig D in S1 Text), which was originally described by Simonsen and Churchill [68]. Due to these irreversible coalescent events, the rate matrix has a block-like structure, and it contains sets of states in which sequences can freely recombine and coalesce until an irreversible coalescent event occurs. Ultimately, both the left and the right sites will have irreversibly coalesced, reaching one of two absorbing states. Note that the matrix is quite sparse, since most of the transitions are not allowed.

Following a similar reasoning, the coalescent with recombination for three sequences can also be modelled as a CTMC. In this case, both the left and the right site will eventually undergo two irreversible coalescent events, which can potentially happen in any order between the three sequences. This creates 203 possible states (Fig G in S1 Text), the transitions of which can also be gathered in a rate matrix (Fig H in S1 Text). This matrix also has a block-like structure, and, given enough time, states will transition into one of the two absorbing states.

If all three sequences belonged to the same species, a three-sequence CTMC would be sufficient to model the coalescent with recombination. However, the sequences belong to three different species, so the speciation process has to be overlaid on top. Subsequently, the coalescent with recombination along the speciation tree is modelled as a series of interconnected CTMCs.

Because sequences are sampled in present time, the left and the right sites are fully linked at time 0, meaning that the starting probability vector for the one-sequence CTMCs is (0, 1). Backwards in time, each of the sequences will remain isolated for a certain period of time in which the two sites can recombine and coalesce freely. The sequences for species A and B will remain isolated until the first speciation event at time $t_A$ and $t_B$, respectively. Then, the final

probabilities of the one-sequence CTMCs for A and B are merged to create the initial probabilities for the two-sequence CTMC. After a certain time, where the two sequences are allowed to coalesce and recombine, the final probabilities for the two-sequence CTMC and the final probabilities for the one-sequence CTMC of species C will be merged, thus creating the starting probabilities for the three-sequence CTMC. Finally, given enough time, all sequences for both the right and the left site will eventually coalesce into one of the two absorbing states of the last CTMC.

## Transition probabilities of the HMM

The hidden states of TRAILS are genealogies which include a topology and two coalescent events that can happen within discretized time intervals. The breakpoints of the time intervals can thus be used to transform the CTMCs into a discrete-time Markov chain (DTMC). First, the joint probability of observing the genealogies and the left and the right loci can be computed by careful bookkeeping of the appropriate paths within the CTMC, defined by the corresponding genealogies and the discretized time intervals. The transition probability matrix of the DTMC (and the HMM) can then be obtained upon dividing the joint probability by the discretized marginals. A detailed description of these derivations is given in S1 Text.

## Emission probabilities of the HMM

For each hidden state, the emission probabilities are calculated using the Jukes-Cantor mutational model [69]. Instead of calculating the emitted nucleotides for the three species only, TRAILS also includes the nucleotides emitted by an outgroup, which provides essential information about the ancestral state in each site. This additional species must have a sufficient divergence with the rest of the species such that ILS can be neglected between them.

## Parameterization

The transition and emission probabilities are parameterized by the speciation times ($t_A$, $t_B$, $t_C$, $t_2$, $t_{upper}$), the effective population sizes ($N_{AB}$, $N_{ABC}$), and the recombination rate (Fig 2A). Implicitly, TRAILS is also parameterized by the mutation rate, but this cannot be jointly inferred with the rest of the parameters because the parameter values can be scaled by any factor and still produce the same coalescent model [57, 70]. Instead, the mutation rate in the model is fixed to 1, and all other parameters are rescaled appropriately. The resulting units for the speciation times are number of generations multiplied by the mutation rate, and, similarly, the effective population sizes are number of individuals times the mutation rate. Accordingly, the recombination rate is divided by the mutation rate, so the optimized parameter is the ratio between the recombination and the mutation rate. After estimating the parameters, parameter values with more interpretable units can be obtained by choosing an appropriate mutation rate.

TRAILS allows for two different parameterizations, namely the *ultrametric* model and the *non-ultrametric* model. In the ultrametric model, all sequences are sampled at time 0, so the molecular clock is assumed ($t_A = t_B = t_C - t_2 = t_1$). Instead, in the non-ultrametric model, each sequence (A, B and C) is allowed to be sampled at a different time. This is useful to model deviations from the molecular clock, for example, when the number of generations for each of the species from present time until the speciation event is different, or the mutation rate varies between the species.

## Simulations

**Parameter estimation.** Simulations to validate the model were performed in `msprime` [29]. The underlying demographic model follows a speciation tree with four species, namely A, B, C and D. The time from present to the first speciation event was set to $t_1 = 200,000$ generations. The (haploid) ancestral $N_e$ for the time between speciation events was set to $N_{AB} = 80,000$. In order to keep an ILS proportion of 32%, the time between the first and the second speciation events was set to $t_2 = -N_{AB} \log\left(\frac{3}{2} \times 0.32\right) = 25,501$ generations. The (haploid) ancestral $N_e$ earlier than the second speciation event was set to $N_{ABC} = 70,000$, and the time between the second speciation event and the speciation event with the outgroup was set to $t_3 = 1,000,000$ generations. The recombination rate was set to $\rho = 0.5 \times 10^{-8}$ per site per generation. The tree was kept ultrametric in number of generations, meaning that all species were sampled at generation 0, so $t_A = t_B = t_1$, $t_C = t_1 + t_2$, and $t_D = t_1 + t_2 + t_3$. Mutations were then added on top of the simulated genealogies according to the Jukes-Cantor model [69] with a mutation rate of $\mu = 1.5 \times 10^{-8}$ per generation per site.

In order to investigate how the number of intervals in the AB-ancestor ($n_{AB}$) and the ABC-ancestor ($n_{ABC}$) affect the parameter estimation, twenty 10-Mb alignments were simulated using `msprime`, and then TRAILS was run to estimate the demographic parameters for $n_{AB} = n_{ABC} = 1$ and for $n_{AB} = n_{ABC} = 5$, using the bound-constrained Nelder-Mead algorithm. The starting parameters of the optimization were randomly drawn from a normal distribution centered on the true value and with a standard deviation of the true value divided by 5. Convergence was achieved at around 150 iterations, with a runtime of around 10 hours for $n_{AB} = n_{ABC} = 5$ per 10-Mb region (see section 9 in S1 Text for further details on the runtime of the model).

**Posterior probability.** The demographic model described above was also used to generate a 100 kb alignment to perform posterior decoding with the true parameters fixed. In TRAILS, the default way of dividing up the coalescent space into discretized time intervals is by taking quantiles of a truncated exponential of rate 1 (measured in units of $N_{AB}$) for the time between speciation events, while the time previous to the earliest speciation event is divided following the quantiles of an exponential of rate 1 (measured in units of $N_{ABC}$). This scheme is appropriate for computing the posterior decoding of the first coalescent event when it happens between speciation events, since it is expected to follow exactly a truncated exponential of rate 1 according to the standard coalescent. However, if the first coalescent happens deep in time, then the coalescent times will follow an exponential with rate 3, since there are 3 lineages present. This means that most coalescent events happen very fast, and there would be an overrepresentation of coalescents in the first interval if the default cutpoint scheme is used.

TRAILS is, however, not restricted to a specific discretization, and it can compute the transition and emission probabilities, and, thus, perform posterior decoding, for user-specified intervals. For the first coalescent in deep time, posterior decoding was performed using cutpoints from the quantiles of an exponential with rate 3, with $n_{ABC} = 7$ and the true parameters from the `msprime` simulation.

Moreover, the second coalescent event will follow a mixture of an exponential with a rate of 1 (for the V0 states) and a convolution of two exponentials with rates 3 and 1 (for the deep coalescent states) [57], which will happen with probability $1 - \exp(-t_2/N_{AB})$ and $\exp(-t_2/N_{AB})$, respectively. Here, $t_2$ is the time between speciation events in number of generations, and $N_{AB}$ is the effective population population size. The second coalescent time can thus be represented as a phase-type distribution [71], with sub-intensity matrix $S$ and initial probability vector $\pi$,

such that

$$S = \begin{pmatrix} -1 & 0 & 0 \\ 0 & -3 & 3 \\ 0 & 0 & -1 \end{pmatrix}$$

and

$$\pi = \left( 1 - \exp\left( -\frac{t_2}{N_{AB}} \right), \exp\left( -\frac{t_2}{N_{AB}} \right), 0 \right).$$

Therefore, posterior decoding for the second coalescent was performed using the quantiles of this phase-type distribution using `PhaseTypeR` [72], with $n_{AB} = 5$, $n_{ABC} = 7$, and the true parameters used to generate the `msprime` simulation.

**Selection.** Using the same demographic model, a 200-kb alignment was simulated using SLiM [30], assuming a single positively selected variant in the middle of the region, with population-scaled selection parameter $2N_e s = 175$. Since SLiM is a forward simulator and runs much slower than backward simulators such as `msprime`, all the demographic parameters of the model were rescaled by a factor of 200 in order to increase computational speed. Posterior decoding was performed on the resulting alignment simulated using the same discretization scheme as described above, and the resulting posterior probabilities are plotted in Fig 4A.

To showcase how the posterior of TRAILS can be used as a test to detect deviations from neutrality, SLiM was used to generate twenty 200-kb with and without a selected variant, and TRAILS was run afterward to calculate the posterior probability for the second coalescent time in 7 discretized intervals. The signal of the posterior decoding was summarized as the mean posterior probability for each discretized time interval, plotted in Fig 4B and 4C.

### Real data

The chromosome 1 multiz alignment of 30 mammalian species (27 primates) was downloaded from the UCSC Genome Browser database in MAF format. Using MafFilter [43], the species of interest were filtered (human, chimp, gorilla and orangutan), syntenic blocks separated by 200 nucleotides or less were merged using human as a reference, and blocks smaller than 2,000 bp were filtered out. The resulting filtered MAF was used as input for TRAILS, using the parameters estimated in Rivas-González et al. [28] as starting values. The optimization was performed using a bound-constrained version of the L-BFGS-B algorithm implemented in `numpy` [45, 46, 73], by setting $n_{AB} = n_{ABC} = 3$, and using the L-BFGS-B algorithm for model fitting. To get a more accurate parameter estimation, the optimized estimates were used as starting values for a second TRAILS run where $n_{ABC} = 5$, optimized using a bound-constrained Nelder-Mead algorithm [74, 75], which showed better convergence for already-optimized TRAILS runs.

Confidence intervals for the estimated parameters were computed using parametric bootstrapping. 20 replicates of 50-Mb regions were simulated from the model fitted with the estimated parameters. Afterward, TRAILS was run on the simulated regions to get optimized parameters. For each parameter, a normal distribution was fitted for the 20 replicates, and the 95% confidence intervals were calculated from the fitted normal (Fig R and Table B in S1 Text).

## Supporting information

**S1 Text. Supplementary notes, including Figs A to S, and Tables A and B.** S1 Text contains a detailed description of the theoretical framework and implementation of TRAILS, together with supplementary analyses.
(PDF)

## Acknowledgments

We gratefully acknowledge Julien Dutheil and Nick Patterson for useful discussions on the implementation of the model. We also thank GenomeDK for providing the computational resources for performing the analyses.

## Author Contributions

**Conceptualization:** Iker Rivas-González, Mikkel H. Schierup, John Wakeley, Asger Hobolth.

**Formal analysis:** Iker Rivas-González.

**Funding acquisition:** Mikkel H. Schierup.

**Investigation:** Iker Rivas-González.

**Methodology:** Iker Rivas-González, John Wakeley, Asger Hobolth.

**Software:** Iker Rivas-González.

**Supervision:** Mikkel H. Schierup, John Wakeley, Asger Hobolth.

**Validation:** Iker Rivas-González.

**Visualization:** Iker Rivas-González.

**Writing – original draft:** Iker Rivas-González.

**Writing – review & editing:** Iker Rivas-González, Mikkel H. Schierup, John Wakeley, Asger Hobolth.

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
