## [Decision Letter · Decision Letter 0]

26 Sep 2023

Dear Dr Rivas-González,

Thank you very much for submitting your Research Article entitled 'TRAILS: tree reconstruction of ancestry using incomplete lineage sorting' to PLOS Genetics.

The manuscript was fully evaluated at the editorial level and by independent peer reviewers. The reviewers appreciated the attention to an important topic but identified some concerns that we ask you address in a revised manuscript.

We therefore ask you to modify the manuscript according to the review recommendations. Your revisions should address the specific points made by each reviewer.

Yours sincerely,

Pier Francesco Palamara

Guest Editor

PLOS Genetics

Xiaofeng Zhu

Section Editor

PLOS Genetics

The reviewers agree that this is a valuable and well-presented approach for parameter inference in multi-species alignment data and provide some suggestions that could improve the manuscript. They agree on the need to provide additional details on the computational costs of TRAILS compared to previous approaches such as Coal-HMM and a discussion on the scalability to additional lineages or time intervals. Additional comments include suggestions for improving the presentation of results, as well as clarifying the effects of deviations from underlying assumptions on the absence of ILS with the outgroup and the role of admixture/introgression/selection.

Reviewer's Responses to Questions

**Comments to the Authors:**

Reviewer #1: Review of "TRAILS: tree reconstruction of ancestry using incomplete lineage sorting" by Iker Rivas-González, Mikkel H. Schierup, John Wakeley, and Asger Hobolth

In this manuscript, the authors present a novel Coalescent Hidden Markov Model framework TRAILS that is geared towards estimating parameters (speciation times & ancestral population sizes) describing the speciation history of closely related species. In particular, the authors present an application to the human-chimp-gorilla scenario, with orangutan as an outgroup. The method takes as input one genomic sequence for each extant species, including the outgroup. The framework implements the marginal genealogical tree relating the extant lineages at each locus as the hidden state space, where the observed nucleotides are the emissions, and the transition probabilities capture the correlation of genealogies at neighboring loci due to chromosomal linkage and recombination. The method can thus take advantage of incomplete-lineage-sorting (ILS) between the extant species to estimate the parameters.

With the novel method, the authors improve upon the method CoalHMM that had previously been presented for similar applications. The new method allows the coalescent events of ancestral lineages between speciation events to be classified into a finer number of discrete intervals, rather than just a single interval. The authors demonstrate that this improves estimation accuracy in simulated data. Applying their method to data from Chromosome 1, they were able to estimate parameter for the speciation times that are in agreement with the literature. In addition to parameter estimation, the authors demonstrate that their method can be used to obtain and inspect the posterior distribution of the marginal genealogies, which they show is useful to investigate ILS along the genome, identify potentially introgressed genomic regions, and identify candidates for adaptive genetic variation.

The manuscript is well written, presenting the novel method and it's application clearly and accessible. The method is well documented and provided as a software package for convenient use. It is thus a valuable addition to the toolkit of methods to characterize speciation events and related phenomena like ILS and introgression, as well as non-neutral dynamics. I do appreciate the detailed supplement and diagrams therein that make the presentation accessible to non-experts. In addition to some minor comments, I think it would be useful to add some additional simulations and analysis to further support the results and expands on some of the applications.

Particular points:

- p.6, Figure 2, Panel B: I do think that displaying the results here as bars that start at 0 is not helpful and unnecessary. It squeezes the whiskers that show the distribution around the mean, making it hard to compare some of them. My suggestion would be to just show a line for the mean of the estimates and whiskers, and the limits of the y-axis chosen to allow better comparison. Perhaps even points for each estimated value from a replicate. Related to this: This plot is based on 5 replicates. While general trends are definitely exposed by this, I do think a higher number of replicates would be better to reduce the random noise.

- p.8, Figure 4: This Figure demonstrates that TRAILS infers low coalescence times around a genetic variant under selection. I think it would be very helpful to supplement this with replicates of simulated 200kb neutral regions, and tally the distribution of the maximum posterior for those. While panel B) shows the theoretical expected value, and deviation from it, It is unclear how much the statistics vary around the mean under neutrality. Knowing this variance is necessary in empirical applications to assess significance of candidate regions. Thus, I think providing some sense of the variability would better exhibit the potential of the method for this application. Please mention here already that the interval boundaries are chosen such that the expected distribution is uniform.

- p.9, l.191-201: In this paragraph, the estimates of the split times are provided as ranges. perhaps confidence intervals. It is unclear how these ranges are computed, since the preceding details describe how to obtain point estimates. Please provide details on whether these ranges are confidence intervals and how they are obtained. Through bootstrapping of the data? Using curvature of the likelihood surface?

- p. 15, l. 357-262: While the details of the method are presented in the supplement, I think it would be good to provide a few more details about the computation of the transition probabilities in this section of the main text. Perhaps add a sentence like: "We can discretize the CTMC into a DMTC by evaluating it at the boundaries of the discretization intervals. This DMTC can the be used to compute discretized joint probabilities of the genealogies (hidden states) at the left and the right locus by considering the corresponding paths of the DMTC. The transition probabilities can then be obtained upon dividing by the discretized marginals."

- The runtime of the method is not mentioned for any of the analyses presented in the paper. To allow researchers interested in applying the method to judge the resources necessary to perform analyses, I think it is necessary to provide more details here. Please provide details on the runtime (and parallel architecture used) of analysing the simulated replicates for Figure 2B), estimating the parameters from the 50Mb HCGO-alignment presented in Figure 5A), and the posterior decoding presented in Figure 5B).

- In the supplement, the state spaces and transition rates for the CTMC with 1,2, and 3 lineages are presented. However, were these obtained by manually enumerating all of them, or is there some structure underlying these that was used by the authors to enumerate them? The reason for this question is that if these are enumerated by hand, extending the method to 4 or more lineages will be very unwieldy, whereas if some structure of the problem can be used, extensions might be less cumbersome. If the authors have some insight into the structure of the problem, and perhaps some more general formulas, please present these.

- The authors do present elegant approaches in the supplement to compute correct probabilities for the CTMC in cases where multiple coalescent events among the ingroup happen in the last "infinite" interval. However, it appears that t_upper is used as an upper bound for coalescent events among the ingroup when computing the emission probabilities. Is this correct? Are these transition and emission probabilities then combined in the HMM? While I do not think that this will majorly effect results if t_upper is large enough, I think this inconsistency should be highlighted (if it does exist).

Minor points:

- p.5, l.112: a bound-constrained search algorithm that optimizes the likelihood function by evaluating it directly. [I think it would be good to state that no gradients or EM are computed.]

- p.5, l.132: Please clarify this statement. Why do these coalescent events cause underestimation?

- p.7, Figure 3, panel A: Please emphasize (perhaps in the caption) that 'first' and 'second' coalescence event refer to the order of events, thus it is possible that different extant lineages are coalescing at this 'first' or 'second' event at different loci.

- p.8, l.169: The signal observed in the posterior decoding can be summarized by comparing the proportion of sites with the maximum posterior probability in certain time intervals to the theoretical expectation.

- p.9, l.186: ..., choosing the parameter values estimated in Rivas-González as starting values ... [it is unclear what "this branch" refers to.]

- p.9, l.188: The supplement states that other algorithms are possible, so state this here too.

- p.9, l.192: Please provide a reference for the value g=25 years.

- p.10, l.225: Wouldn't it be more appropriate to have a time in years represent each interval and then take the weighted mean of that?

- p.16, l.404: It is stated here that the Nelder-Mead algorithm is used for optimization. Previously, it was stated that the L-BFGS-B algorithm is used. Please clarify.

- p.16, l.409: ... posterior decoding with the true parameters fixed.

Supplement:

- p.2, l.55: ... and sit in different lineages.

- p.4, Figure S3: Add to caption: "Grey indicates the diagonal entries, which are computed as the negative of the sum of the off-diagonal entries in the corresponding row."

- p.4, l.94: Please provide more details how the probabilities are mixed.

- p.6, l.112: ... point t using \\pi_{ABC}' = \\pi_{ABC} exp(tQ_{ABC}).

- p.5, l.115: Remove one period.

- p.8, l.125: ... two topologies is known as incomplete lineage sorting ...

- p.9, l.141: Why is the number given by the Bell number series? Please provide an explanation or a citation.

- p.11, l.181: ... two-sequence CTMC, and, later, that lineage coalesces with ...

- p.12, l.187: Additionally, if the first coalescent event does not happen between ...

- p.16, l.286: Does this need to be F(t) = e^{tQ}?

- p.16, l.293 (and following equations): I believe the order of the matrix exponentials has to be reversed? Each next step has to be multiplied from the right. Thus, e^{rQ} should be the leftmost exponential, followed by (s-r), followed by (t-s). Similar with most other equations in the following sections. I might also be wrong about this.

- p.17, l.325: ... we need to calculate infinite integrals of ...

- p.19, l.343: The states of the DTMC describe the marginal genealogical histories of the sequences. However, these states cannot be observed directly.

- p.21, l.378: Should the rate of the exponential be the inverse of the ancestral population size instead of 1?

- p.23, l.417: Pr(a_0) is not defined. Is it the stationary distribution of the mutation matrix?

Reviewer #2: This is a nice paper and I enjoyed reading and thinking about it. The authors extend a previous approach to ancestral demographic inference from a multi-species genome sequence alignment, by introducing a more sophisticated representation of the coalescent process.

I don't have too many comments or suggestions to make, as it's a fairly self-contained methodological study and the manuscript motivates and describes the methods and approach well. I'm persuaded that this is a useful approach, and a potentially powerful framework for tackling problems in this area. The results on the great ape alignment provide a helpful demonstration of how it might be used.

There were just a few things I think the authors might address in a bit more detail, two of which relate to assumptions of the model.

Firstly, the model explicitly assumes that the outgroup is sufficiently remote that there is zero ILS between it and the A, B and C lineages involved in the focal speciation events. But at the same time it assumes that e.g. mutation and recombination rates have remained unchanged on all these lineages. In reality neither assumption might hold. But the ILS assumption seems particularly relevant. For example in the case of the HCGO divergence there will be about 13% ILS between HC, G and O using the parameters estimated in the paper. How does this impact the performance of the method or inferences drawn from it? Can it be mitigated in filtering the input alignment blocks? (I didn't see a discussion of this in the Methods.)

Secondly, TRAILS fits a 'clean split' speciation model in which there is no admixture between branches after their divergence. The authors discuss how the method might respond when there is potential departure from this in the data, in the context of the long V3 fragment in Fig. 5. One question which arises is whether there are more systematic approaches to detect such signals. For example, can one identify or quantify unexpectedly long fragments based e.g. on their posterior odds under the HMM? For having identified them, one could then look at the numbers of V3 and V2 topologies. Genome-wide asymmetry in these classes could be diagnostic of admixture or introgression, whereas the effects of selection if widespread might be expected to be symmetric. Did (or might) the authors investigate this?

Thirdly, I think it is fair to say that the method is considerably more complex in terms of its underlying machinery than previous approaches such as CoalHMM. It would be good if the authors could comment on how this influences performance and scaling considerations, e.g. what are typical run times, memory requirements etc for the cases presented, particularly as one adds time intervals?

I noticed a couple of typos, in the equation on p. 9 and the preceding text, N_{ABC} should surely be N_{AB}.

I liked the Supplement a lot; it provides a clear discussion which addressed most of the questions I had about the method and its implementation. I do think it will still be difficult to follow for anyone new to the ideas involved, but that's perhaps unavoidable. I have a couple of minor suggestions.

In discussing the basic CTMC, I wonder is it worth noting that there are two aspects of coalescence involved - one being the merging of homologous sequences at a particular locus (e.g. in going from \\omega_{00} to \\omega_{30} or \\omega_{03}), and the other being the linking of two separate loci (e.g. in going from state 1 to another state in \\omega_{00}). It's a minor thing but I think many readers will be more familiar with the first than the second. You might also consider reproducing the state diagram for the CTMC in the single-sequence case, to illustrate the process.

The other suggestion is to change the red/blue colours in Fig S5, as there is potential confusion with other figures in which the same colours distinguish sequences/species.

Aylwyn Scally

Reviewer #3: The ms by Rivas-Gonzalez et al. reports a novel powerful extension of the Coal-HMM framework published by part of the authors some years ago. The ms also advertises for TRAILS, the newest release of a series of softwares that infer ancestral population sizes, speciation times and recombination rates for a genomic alignment of 3 species, plus an outgroup. The study assesses the power and the limitations of the method (and the related software) with great care, using simulations and a human-chimp-gorilla+orang-utan alignment.

I would like first to thank the authors for the care they took to explain the method in a very clear and comprehensive supplementary material. With only basic knowledge of HMM and coalescent theory, it is quite easy to follow, enlightening and enjoyable. It helped me a lot having a better grasp of what was at stake and improved a lot my comprehension of Coal-HMM techniques. Thank you.

More generally, the ms is scientifically sound, easy to read and quite convincing. I have only a list of comments and suggestions that may help to produce an even better/clearer article.

1) My first and most important suggestion is to change your strategy for the figures 2-5 of the main text. As they are currently, they are quite difficult to read and even more to understand. I suspect that you were tempted to provide as much info as you could, but in the end, a casual reader such as my poor self can suffer from the impression of being overwhelmed by the generosity of the figures. I shall now detail few more concrete suggestions, figure by figure.

- Figure 2. Panel A, what is t_upper? Wouldn't a t_3 ranging from AB to ABC be more intuitive? Or maybe there is something I don't get (probably). On panel B, I am not sure whether having both n_{AB}=3 and n_{AB}=3 is really helping. Reducing the number of subplot can only make the figure clearer. In the current format, it is too crowded and fonts are too small for the readers. Furthermore, why don't you use whisker-plots instead of bar-plots?

- Figure 3. I wonder whether the names "V0" to "V3" is a better choice than newick self-explanatory strings such as "((A,B),C)"? Furthermore the posterior probabilities within the heatmaps are not well contrasted so it is not visually convincing that the HMM does a good job (with the exception of the topology). Did you try using log(prob) for the color code? or less categories ? Furthermore, the tree with tiny slices denoted by Ss and Fs is way to small to be read and again not very helpful. Please make it more straightfoward.

- Figure 4. Same remark for the colors of heatmaps. It is not obvious how the Posterior max is computed. Finally, theoretical expectations means theoretical NEUTRAL expectations, correct?

- Figure 5. Panel B has remained opaque to me (I have abandoned, even as a reviewer). Here again, recalling what "V0" and others are implies going back and forth between Figure 5 and Figure 2. I again believe the newick strings is a better choice than V[0-3]. Like in general, font sizes and plots are too small.

2) The underestimation of rho (figure 2 and l131) is intriguing. As the convergence of the other estimates is good, or even very good, this is mind-bugging. The authors suggest that it stems from rapid coalescence after recombination that results in undetectable recombination events. But really what intuitively matters is the occurrence of mutations in the time lag between both types of events. In this case, tuning the \\mu to \\rho ratio would change the strength of the bias, lowering it as it increase. More generally, as it is the only poor estimate, I recommend explore different strategies to overcome the bias or at least better characterizing the issue and discussing it more.

3) The authors provide an estimate of the parameter from a single region of chromosome 1. Having few regions from the same and from different chromosomes and comparing the results is certainly a good move. I somehow have a vague memory that one of the chromosomes had a different pattern of ILS, but I may be totally wrong (this an old memory).

4) Reports of CPU time and memory consumption are lacking. Especially discussing them regarding the differences with previous simpler versions. In general, it is interesting to know how much CPU resource (e.g. carbon) we spend for how much precision we gained. What did we gain for what cost? About CPU time, the complexity is likely linear with alignment size, but how is it with number of time categories in AB and in ABC.

:: A collection of minors remarks ::

l163 : this approximation is not very good. A better one is "2ln(2Ns + c)/s" where c is the Euler constant. At least better use 2Ns than 2N.

l199: insert "new"  "our NEW estimates"

l269-l278: can we see the xy-plots of the correlated variables (probably in the supp)?

- figure 5C and l257-291. Any temptation to compute some kind of standard neutrality test (e.g. dn/ds on the coding, branch length inflation, or SFS-based using publicly available polymorphisms for this locus)?

- l 321 & l56 (supp): please state that the coalescent occurs between the two ancestors that were carrying the left and right ancestral materials. I did get it, but at first I was disturbed.

- l368 : "ILS can be neglected" instead of "no ILS".

- l429: the second "forward" should be replaced by "backward"

- l112 (supp) \\pi_{AB} should be \\pi_{ABC}

- l115 (supp) ".." -> "."

- p9-10 (supp). I guess the matrices were recoded in sparse format, which really speed up calculation and reduce memory.

- l286 (supp). I am not sure but I think it should be exp(tQ) and not exp(tA)

- l473. This "e" is different from the (2,1) vector described earlier l265, no?

- l481. It is not obvious to me how it works as B is a matrix and e a vector (from what I can understand).

**Have all data underlying the figures and results presented in the manuscript been provided?**

Reviewer #1: Yes

Reviewer #2: Yes

Reviewer #3: Yes

PLOS authors have the option to publish the peer review history of their article (what does this mean?). If published, this will include your full peer review and any attached files.

Reviewer #1: No

Reviewer #2: **Yes: **Aylwyn Scally

Reviewer #3: No

---

## [Decision Letter · Decision Letter 1]

22 Jan 2024

Dear Dr Rivas-González,

We are pleased to inform you that your manuscript entitled "TRAILS: tree reconstruction of ancestry using incomplete lineage sorting" has been editorially accepted for publication in PLOS Genetics. Congratulations!

Yours sincerely,

Pier Francesco Palamara

Guest Editor

PLOS Genetics

Xiaofeng Zhu

Section Editor

PLOS Genetics

Comments from the reviewers (if applicable):

All major comments have been addressed. Reviewer 1 has a few minor suggestions that the authors may want to consider.

Reviewer's Responses to Questions

**Comments to the Authors:**

Reviewer #1: Second review of "TRAILS: tree reconstruction of ancestry using incomplete lineage sorting" by Iker Rivas-González, Mikkel H. Schierup, John Wakeley, and Asger Hobolth

The authors have addressed my major concerns. I do believe that the additional simulations to assess the variability of the parameter estimates (and revised Figure 2) and the posterior decoding allow the reader to better assess the method and how it performs in applications. The added discussion of the runtimes further helps. I do have two minor suggestions, which, I think, can be left to the authors to potentially address. I don't think that addressing them is necessary for publication.

- It could be good to refer to the supplmentary section "Beyond three species" in the "Discussion" in the main text as potential for extensions of the method.

- I believe that the coalescence rates, recombination rates, and mutation rates in the computations presented in the supplement are all scaled by N_e? If so, it might be worthwhile mentioning this, as it also affects the boundaries of the discretization intervals and the interpretation of the inferred times (see comment and response on t_upper from Reviewer #3).

Reviewer #2: These edits look good to me and I'm satisfied that the authors have addressed the points I raised.

Reviewer #3: I congratulate the authors to have edited their ms to make it even more clear and sound. I have no further comment.

**Have all data underlying the figures and results presented in the manuscript been provided?**

Reviewer #1: Yes

Reviewer #2: Yes

Reviewer #3: Yes

PLOS authors have the option to publish the peer review history of their article (what does this mean?). If published, this will include your full peer review and any attached files.

Reviewer #1: No

Reviewer #2: **Yes: **Aylwyn Scally

Reviewer #3: No

**Data Deposition**

http://datadryad.org/submit?journalID=pgenetics&manu=PGENETICS-D-23-00699R1

**Press Queries**

---

## [Editor Report · Acceptance letter]

5 Feb 2024

PGENETICS-D-23-00699R1 

TRAILS: tree reconstruction of ancestry using incomplete lineage sorting 

Dear Dr Rivas-González, 

We are pleased to inform you that your manuscript entitled "TRAILS: tree reconstruction of ancestry using incomplete lineage sorting" has been formally accepted for publication in PLOS Genetics! Your manuscript is now with our production department and you will be notified of the publication date in due course.

With kind regards,

Judit Kozma

PLOS Genetics

On behalf of:
